# Talking Models: Distill Pre-trained Knowledge to Downstream Models via Interactive Communication

## Abstract

Many recent breakthroughs in machine learning have been enabled by the pre-trained foundation models. By scaling up model parameters, training data, and computation resources, foundation models have significantly advanced the state-of-the-art in many applications. However, it is still an open question of how to use these models to perform downstream tasks efficiently. Knowledge distillation (KD) has been explored to tackle this challenge. KD is a technique that transfers knowledge from a large teacher model to a smaller student model. While KD has been successful in improving student model performance, recent research has discovered that a powerful teacher does not necessarily lead to a powerful student, due to their huge capacity gap. In addition, the potential distribution shifts between the pre-training data and downstream tasks can make knowledge transfer in KD sub-optimal for improving downstream task performance.

In this paper, we extend the knowledge distillation paradigm by introducing an interactive communication process to help student models of downstream tasks learn effectively from pre-trained foundation models. Our design is inspired by the way humans learn from teachers who can explain knowledge in a way that meets the students' needs. Specifically, we let each model (i.e., student and teacher) train two components: (1) an encoder which encodes the model's hidden states to a message in a shared message space and (2) a decoder which decodes any message to its own hidden states. With encoder and decoder, not only can the teacher model transfer rich information by encoding its hidden states to messages, but also the student model can send messages with information of downstream tasks to teacher so that the teacher can interpret and generate responses. With this interactive communication process, knowledge passing from teacher to student can be tailored to the student's model capacity and downstream tasks' distributions. We conducted experiments on benchmark datasets for computer vision and recommendation tasks to show that our communication mechanism outperforms state-of-the-art distillation techniques.

## 1 Introduction

Scaling up machine learning models has been shown to improve performance in many applications, including Natural Language Processing (NLP) (Chowdhery et al., 2022), Computer Vision (CV) (Yuan et al., 2021) and Information Retrieval tasks (Tay et al., 2022). For example, the number of parameters of newly developed language models for NLP tasks has grown from hundreds of millions to hundreds of billions in the past few years (Zhao et al., 2023). By scaling both model size and training data, researchers have found that not only can large models reduce training loss (Kaplan et al., 2020), they can also show emergent abilities to solve tasks that smaller models cannot solve, such as in-context learning and instruction following (Wei et al., 2022). Similarly, in CV, large models can improve performance of multiple tasks of different datasets (Dosovitskiy et al., 2020). The training of these large models can be extremely costly, but once they are trained, they can be stored as pre-trained foundation models to be fine-tuned for downstream tasks (Raffel et al., 2020; Ridnik et al., 2021).

As the size of these foundation models grows bigger, the inference cost of the fine-tuned models also grows. On the contrary, for many real-world machine learning applications, researchers have built

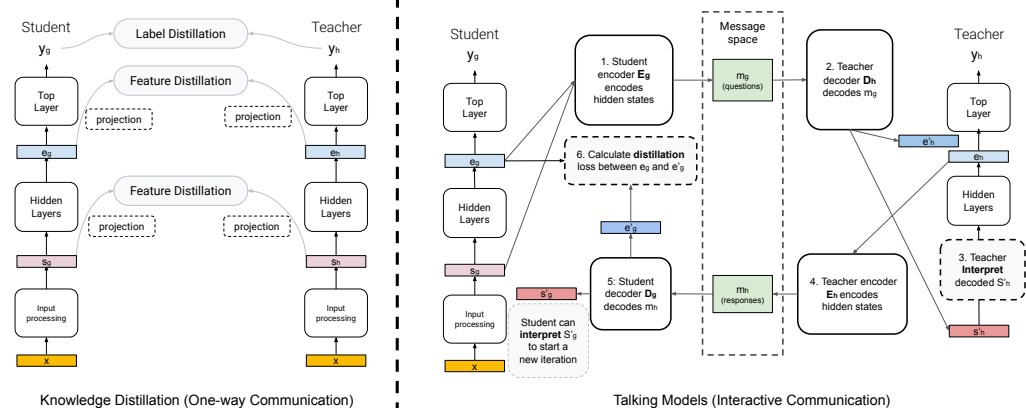

Figure 1: Talking-model Distillation (TD, right) compared to Knowledge Distillation (left, which can be label distillation (Hinton et al., 2015), feature distillation (Heo et al., 2019; Romero et al., 2014) or hybrid of both (Zhu & Wang, 2021)). In TD, student $M_g$ encodes its hidden states given input $x$ to message $m_g$ with $E_g$, teacher $M_h$ then decodes it with $D_h$, and interprets the decoded hidden states with its hidden layers. $M_h$ encodes the interpreted states to message $m_h$ with $E_h$, Then $M_g$ decodes it with $D_g$ and use it for distillation. $M_g$ can also interpret decoded hidden states and generates "follow-up" messages to communicate for multiple iterations.

their own sophisticated but efficient models, such as real-time traffic sign detection (Bousarhane et al., 2021), or recommendation and retrieval (Zhao et al., 2019). Without techniques that can significantly bring down the inference cost of the fine-tuned models, it is challenging to apply foundation models to a wide range of applications with extensive inference requests and strict constrains on latency.

Existing efforts in reducing the inference cost of large machine learning models include model pruning (Frantar & Alistarh, 2022), quantization (Liang et al., 2021) and knowledge distillation (Hinton et al., 2015). Model pruning (Frantar & Alistarh, 2022) techniques prune learned weights based on their importance, and quantization (Dettmers et al., 2022) techniques approximate learned weights with data-types of low-bit-width. While both techniques provide a trade-off between efficiency and quality, they either need to be carefully designed for specific model architectures, or would degrade the quality of the model regardless of the downstream tasks. When transferring knowledge of pre-trained foundation models to perform downstream tasks, it is desirable to have flexibility for the downstream models adopting different model architectures, such as decoders from the pre-trained model. Therefore, knowledge distillation techniques, which transfer knowledge from one model to another by using model outputs (e.g., predictions), can provide such flexibility (Gou et al., 2021). In this paper, we explore the direction of distilling the knowledge from pre-trained foundation models (teacher) to smaller models for downstream tasks (students) by knowledge distillation.

Some very recent research (Hsieh et al., 2023) successfully distilled large language model (LLM) to smaller models for downstream tasks, by using LLM generating rationale of the labels for the downstream models. As this research is specific to NLP applications and cannot be easily extended to other machine learning applications, these exist two additional challenges on applying KD to distill pre-trained knowledge to improve downstream models. Firstly, recent research discovers that a huge capacity gap between teacher and student will hurt KD (Wang et al., 2022; Huang et al., 2022). Therefore it becomes difficult to apply KD to improve smaller downstream models by learning from much larger pre-trained models. Secondly, downstream tasks can have different distributions from the pre-trained data. Although some recent study found that distillation from a fine-tuned teacher can overcome distribution differences and improve the performance of the student model (Wu et al., 2023), fine-tuning pre-trained foundation models is very costly and cannot scale to many downstream tasks. Without full fine-tuning, knowledge from the pre-trained model may not be directly useful to downstream tasks.

We hypothesize that the challenges arise when the pre-trained teacher do not have a good understanding of the student's capacity and ability in learning downstream tasks. We view KD as a one-way communication process between the teacher and student: the teacher generates information (e.g., logits, hidden states) from training data and passes it to the student. And one-way communication can be sub-optimal for knowledge transfer since teacher has no information about the student. In this paper, we extend the KD framework by introducing a two-way communication process, where

teacher can provide specific guidance by interpreting student's request. This is inspired by how human communicate and learn from each other: persons from different background and expertise can exchange knowledge effectively by interactive conversations; and students ask question and learn from teacher's response and continue asking follow-up questions. This can be formulated as the Osgood and Schramm Model of communication (Wrench et al., 2020) (shown in Figure 2), which is an interpersonal interaction model indicating that messages can go in two directions with a heightened focus on cyclical feedback.

We adapt the Osgood and Schramm Model to design the interactive communication process for KD, where students can ask questions to teachers and learn from the answers. Our proposed process, named Talking-model Distillation (TD), is shown in Figure 1 (on the right, compared to KD on the left). To be more specific, for each model (i.e., the teacher or student), we introduce two modules used for communication: an encoder that encodes model's hidden states to a message, which will be passed to the other model; and a decoder that decodes the message to its hidden states. The communication process starts with student model encoding its hidden states given an input to a message. Teacher then decodes the message into hidden states in its own hidden space. We introduce an interpreting step where teacher runs the decoded hidden states (for bottom layers) using its own learned weights to generate new hidden states. Teacher then encodes the interpreted hidden states into message and returns it to student. Student uses the returned message for knowledge distillation by aligning the decoded hidden states in the returned message with its own hidden states. The training of the encoder and decoder for both teacher and student can happen at the same time as training the student, while the learned weights of the teacher will not be updated. Given the encoder and decoder for each model can be as simple as projection layers using a small number of parameters just to align the corresponding hidden space to a shared message space, the learning of communication parameters does not add much overhead in training the student model.

By introducing this interactive communication process, the student can request information from teacher based on its own hidden states generated from the downstream tasks. The information teacher provides through the message space is better aligned with downstream tasks by both models' encoders and decoders. Moreover, student can also interpret teacher's returned message and generate a new message back to teacher, as a follow-up. This enables the cyclical communication behavior where multiple iterations of communication can happen, so that student can get sufficient tailored guidance from teacher, even when downstream tasks have extremely sparse training data (e.g., movie recommendation for uncommon genres, studied in our experiments).

To show that our proposed method can be used in different domains and applications, we conduct experiments on multiple benchmark datasets including computer vision tasks and recommendation tasks. Compared with state-of-the-art distillation methods (e.g., a hybrid of feature and label distillation (Zhu & Wang, 2021)), our interactive communication process significantly improves knowledge transfer between pre-trained teacher and students of downstream tasks.

## 2 RELATED WORK

In this paper, instead of improving general KD, we focus on a specific use-case: use pre-trained large models to teach smaller students for downstream tasks, without extensive fine-tuning of the teacher. Therefore, many most recent KD algorithms cannot be directly applied, as they assume similar tasks between teachers and students (Yang et al., 2022b; Beyer et al., 2022). In this section, we discuss existing research for efficient model tuning, including model compression and KD, and their inspirations to improve efficiency for downstream models in the pre-training fine-tuning paradigm.

### 2.1 MODEL COMPRESSION FOR EFFICIENT SERVING

There exist many widely studied research directions to reduce the serving cost of deep neural networks, including but not limited to model pruning (Hassibi et al., 1993), quantization (Guo, 2018), and knowledge distillation (Hinton et al., 2015). Recent model pruning methods (Frantar & Alistarh, 2022; El Halabi et al., 2022; Kwon et al., 2022) have considered large pre-trained models and potential sparse downstream tasks. In these methods, sub-modules of the pre-trained models can be pruned based on their importance towards a pre-defined metric (Frantar & Alistarh, 2022; El Halabi et al., 2022), or masks can be learned to mask out learned weights given downstream tasks (Kwon

et al., 2022). While pruning methods need to be designed based on different model architectures, quantization (Krishnamoorthi, 2018; Guo, 2018) can be applied to any learned weights by replacing them with low-precision data-types.

Both pruning and quatization directly modify the learned weights of pre-trained models, however, a model with a different structure can be desirable for downstream tasks. Both techniques cannot be applied to newly initialized weights. Knowledge Distillation provides such flexibility and can be complimentary to be used together with pruning and quatization methods (Shin et al., 2019).

## 2.2 KNOWLEDGE DISTILLATION

Knowledge Distillation uses knowledge from a larger, more powerful "teacher" model to improve the performance of a smaller, more efficient "student" model (Gou et al., 2021). In recent years, KD has been used in many applications, such as NLP (Sun et al., 2019; Sanh et al., 2019), CV (Park et al., 2019; Tung & Mori, 2019), and Recommendation Systems (Tang & Wang, 2018).

However, recent research discoveries in understanding knowledge distillation (Wang et al., 2022; Huang et al., 2022; Zhu et al., 2022) suggest a larger teacher does not necessarily guarantee a better student. On the contrary, a huge capacity gap between teachers and students can lead to little or no improvement for the students. This is because some specific representations can only be learned with a large capacity model (Zhu et al., 2022), or an information bottleneck can be created due to the capacity gap (Wang et al., 2022). This is especially challenging when distilling knowledge from large pre-trained foundation models.

To deal with the capacity gap, existing works introduce intermediate steps/models such as teaching assistant models (Mirzadeh et al., 2020). Additionally, they try to leverage more information from teachers besides their predictions (or logits), such as feature representations (Heo et al., 2019; Yang et al., 2019; Zhu & Wang, 2021; Romero et al., 2014), relationships among labels and representations (Hao et al., 2022; Huang et al., 2022), or the weights of teacher models (Fu et al., 2021).

In this paper, we discuss a specific case of distilling pre-trained foundation models for downstream tasks, where downstream tasks can be different from pre-trained tasks. Fine-tuning a pre-trained foundation model for each downstream task to obtain a specific teacher model can be costly and cannot scale up to many different downstream tasks. Existing research in cross-task distillation (Yang et al., 2022a; Clark et al., 2019; Zhong et al., 2022) shows that a teacher model trained from multiple tasks can be potentially helpful to a single task student (Clark et al., 2019). However, for downstream tasks which are not present in teacher models' pre-trained tasks, auxiliary tasks will be needed for both the teacher and student (Yang et al., 2022a). Therefore, we focus on directly distilling information from pre-trained models to students, using their feature representations in hidden spaces similar to the feature distillation techniques (Zhu & Wang, 2021).

## 2.3 IMPROVE FOUNDATION MODEL EFFICIENCY FOR DOWNSTREAM TASKS

The successes of large pre-trained foundation models inspire researchers to explore techniques to improve the efficiency of using them in real-world applications. Many existing research has been focusing on how to improve the efficiency in training or fine-tuning the foundation models. This includes adaptor based method (Houlsby et al., 2019), which freezes most of the learned weights and only trains a small adaptor layer for each downstream task. More recently, parameter-efficient fine-tuning techniques extend adaptors to low-rank adaptors which can further improve the training efficiency while reducing serving latency (Hu et al., 2021; He et al., 2022). However, the serving cost of adaptor based fine-tuned model is still comparable to directly using the foundation models.

More recently, there were early empirical results (Hsieh et al., 2023; Wu et al., 2023) showing that distillation method can be used to distill knowledge from pre-trained foundation models to downstream tasks. In NLP specific tasks, by using pre-trained model to generate rationales and explanations of downstream tasks labels, more information can be passed from teacher to student to improve knowledge transfer (Hsieh et al., 2023). For text-image foundation model, empirical results showed that vanilla distillation might not directly improve downstream task unless a fine-tuned teacher is used (Wu et al., 2023). Inspired by their discoveries, we target at developing distillation techniques that don't require fine-tuning to downstream tasks and can be applied to different machine learning applications and downstream tasks such as CV and Recommendation Systems.

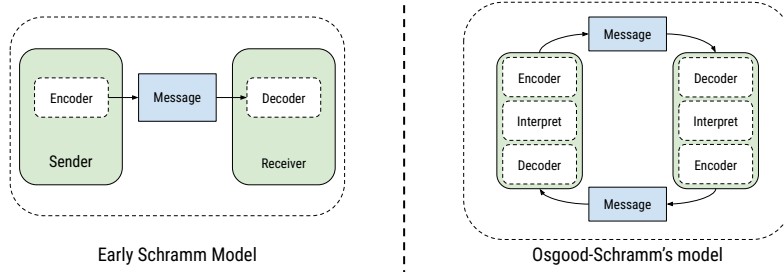

Figure 2: The interpersonal communication models. One-way communication model (left, Early Schramm Model) and interactive communication model (right, Osgood-Schramm's Model)

## 3 PROPOSED METHOD

In this section, we introduce our proposed method named Talking-model Distillation (TD). Our key innovation is to generalize the existing KD methods as a one-way communication process and extend it to a two-way interactive communication process. We first describe how to adapt interpersonal human communication process to machine learning models for KD between teacher and student. We then show that existing KD can be categorized as a one-way communication process. Then we will introduce how TD enables the teacher and student to communicate interactively so that the student can learn rich information from the teacher with specific focus on downstream tasks.

### 3.1 COMMUNICATION ENCODER AND DECODER

In Figure 2, we show two interpersonal communication models in communication theory (Wrench et al., 2020), named Early Schramm Model and Osgood-Schramm's Model. Early Schramm Model is a one-way communication model where a sender encodes a message and passes it to a receiver to decode. The Osgood-Schramm's Model is an interactive communication model that two individuals can send, receive and parse messages in a cyclical way, so that the messages being passed can capture feed-backs from each other.

The key components for both communication processes are: (1) Encoder that encodes information to communicate as messages; (2) Decoder that decodes message to understandable information; (3) A shared message space. To apply these communication models to knowledge distillation, for each of the teacher and student models, we introduce encoder and decoder for communication.

The first question is what information can encoder encode, while the original KD used logits (Hinton et al., 2015), we build encoder on top of the hidden representations of the model, which are more flexible and are not tied to the final prediction tasks (Heo et al., 2019). Given a model $M_g$ which generates predictions $y_g$ for labels $y$ given input $x$, it typically has multiple hidden layers $H_1^g, ... H_{n_g}^g$ (e.g., Multi-layer perception (Gardner & Dorling, 1998), Convolution layers (Yamashita et al., 2018), or Transformer layers (Vaswani et al., 2017)). The encoder can encode any hidden states from these layers to messages. Without loss of generality, we categorize the hidden states into two categories: (1) lower level hidden states $s_g$, which capture the lower-level representations or processed input signals from $H_1^g$ to $H_{l_g}^g$, and (2) higher level hidden states $e_g$, which capture the embeddings that can be used for a few top hidden layers from $H_{l_g+1}^g$ to $H_{h_g}^g$ to generate predictions. These hidden layers for $M_g$ to generate predictions $y_g$ given $x$ can be written as:

$$y_g = H_{h_g+1,...,n_g}^g(e_g) = H_{h_g+1,...,n_g}^g(H_{l_g+1,...,h_g}^g(s_g)) = H_{h_g+1,...,n_g}^g(H_{l_g+1,...,h_g}^g(H_{1,...l_g}^g(x)))$$

Then the encoder $E_g$ for model $M_g$ encodes hidden states from both lower and higher levels into message $m_g$ in the message space $m_g = E_g(\{s_g; e_g\})$. Therefore another model $M_h$ can use its decoder $D_h$ to decode $m_g$ into hidden states for $M_h$: $\{s_h'; e_h'\} = D_h(m_g)$.

The selection of communication encoder and decoder's structure depends on the model architecture as well as the design of the message space. For example, it can be Transformer layers if the model is a transformer and message space contains sequences of embeddings. In this paper, for simplicity, we only explored linear or Dense-Relu-Dense encoders and decoders with message space of $m \in \mathbf{R}^{m_d}$, where $m_d$ is the dimensionality of the message space.

## 3.2 EXISTING KD TECHNIQUES AS ONE-WAY COMMUNICATION

Next we show how we can formulate existing KD techniques as a one-way communication process with the encoder and decoder introduced in the previous subsection. KD techniques here can refer to logit distillation (Hinton et al., 2015), feature distillation (Romero et al., 2014; Heo et al., 2019) or a hybrid of both (Zhu & Wang, 2021). We denote student model as $M_g$ and teacher model as $M_h$

For logits distillation, we can simply view the logit space as the message space given both teacher and student having the same prediction tasks. And the distillation loss is $L_{logit} = d(y_g, y_h)$, where d is a distance metric such as L2 loss or KL divergence between the teacher and student logits.

For feature distillation method (such as (Heo et al., 2019)) which projects hidden states of different dimensionalities into a shared space, i.e., the shared message space, the distillation loss can be calculated by distance measurement between messages in the message space.

$$L_{feature} = d(m_g, m_h) = d(E_g(\{s_g; e_g\}), E_h(\{s_h; e_h\})) =$$

$$d(E_g(\{H^g_{1,\ldots l_g}(x); H^g_{l_g+1,\ldots,h_g}(H^g_{1,\ldots l_g}(x))\}), E_h(\{H^h_{1,\ldots l_h}(x); H^h_{l_h+1,\ldots,h_h}(H^h_{1,\ldots l_h}(x))\}))$$

Here $d$ is the distance between two messages, which can be L2, KL-divergence or any of the relational metrics that have shown promising results for feature distillation such as Pearson correlation (Huang et al., 2022) or manifold loss (Hao et al., 2022). Encoders for $E_g$ and $E_h$ are learned parameters for feature distillation.

Another slightly different feature distillation method, Fitnet (Romero et al., 2014), which projects teacher model hidden states to student model's hidden states, can be viewed as student model $M_g$ directly decoding teacher model $M_h$'s hidden states (as message, where $E_h$ is an identity transformation), hence the KD loss can be written as:

$$L_{fitnet} = d(\{s_g; e_g\}, D_g(m_h)) = d(\{s_g; e_g\}, D_g(\{s_h; e_h\})) =$$

$$d(\{H^g_{1,\ldots l_g}(x); H^g_{l_g+1,\ldots,h_g}(H^g_{1,\ldots l_g}(x))\}, D_g(\{H^h_{1,\ldots l_h}(x); H^h_{l_h+1,\ldots,h_h}(H^h_{1,\ldots l_h}(x))\}))$$

where $D_g$'s parameters are learned. We can see that $L_{fitnet}$ distills knowledge by minimizing the distance between original student model's hidden states and decoded states from teacher's hidden states. And Feature distillation $L_{feature}$ distills knowledge by minimizing the distance between encoded student's hidden states and teacher's hidden states in a shared message space.

## 3.3 INTERACTIVE COMMUNICATION

Our proposed method, named Talking-model Distillation (TD) uses interactive communication process, which allow both teacher and student model to interpret messages and return new messages. To be more specific, below we describe one iteration of such interactive communication:

**Student passes message to teacher** Student model $M_g$ generates a message $m_g = E_g(\{s_g; e_g\})$ from input $x$ and pass it to teacher model $M_h$. Then teacher model decodes the message into hidden states with its decoder $\{s'_h; e'_h\} = D_h(m_g)$.

**Teacher interprets message and encodes returned message** We introduce an **interpreting** step, where teacher model interprets the decoded message by its own learned weights. Teacher model uses the decoded lower-level hidden states $s'_h$ as input to its hidden layers $H^h_{l_h+1}, ..., H^h_{h_h}$ to generate interpreted states $\tilde{e}_h = H^h_{l_h+1,\ldots,h_h}(s'_h)$. Then teacher encodes the interpreted states (along with the decoded lower-level hidden states) as a returned message: $m_h = E_h(\{s'_h, \tilde{e}_h\})$. The interpreting step is crucial for the interactive communication process, as it enables messages from teacher encodes information of teacher's knowledge (model parameters) being applied on student's messages.

**Student decodes returned message and learns from teacher** The student model $M_g$ decodes the returned message to its hidden space: $\{s'_g; e'_g\} = D_g(m_h)$. Then student can learn from teacher by minimizing the distance of the decoded states with its original states:

$$L_{interact} = d(\{s_g; e_g\}, \{s'_g; e'_g\}) = d(\{s_g; e_g\}, D_g(E_h(\{s'_h; \tilde{e}_h\})))$$

Here $d$ can be any distance metric used in existing feature distillation techniques. In this paper, we use L2 loss as $d$ for all distillation methods. We can see the key differences between $L_{interact}$ and other

distillation losses ($L_{logit}$, $L_{feature}$, and $L_{fitnet}$), is that we apply teacher's hidden layers and learned weights on interpreted student's messages, instead of input $x$. By doing so, along with the training of both models' encoder and decoder, teacher can provide feedbacks that fit student model's capacity and learned representation space. Note that $e'_h$ is not used to calculate $L_{interact}$ for learning from the teacher. But it is used for training the teacher's decoder in $L_{SC}$ discussed in the next subsection.

**Student interprets and generates follow-up messages**    After receiving the returned message from teacher, student model can also interpret the message $m_h$ and generate an interpreted state: $\tilde{e}_g = H^g_{l_g+1,\ldots,h_g}(s'_g)$. Then student model can encode it with $s'_g$ to $m_h^2 = E_g(\{s'_g, \tilde{e}_g\})$. Here $m_h^2$ refers to the second iteration of the message for interactive communication. It will be passed to teacher again to start the next iteration of communication.

Note that new iteration of communication won't consume new input $x$ and this can continue for as many iterations as possible. By doing so, rich information from teacher model can pass to student based on student's request, even when downstream tasks are extremely sparse. The iterative update scheme here shares similarity to some techniques in Semi-Supervised Learning and Self-Training (Xie et al., 2020), however, our proposed method used in knowledge distillation are more light-weight and only updates the student model with iterations of interactive communications. We explore the number of iterations for interactive communication as a hyper-parameter in our experiment.

To sum up, our proposed communication process is shown in Figure 1, compared with other KD methods. By comparing Figure 1 with Figure 2, we can see that the differences between our method and other KD techniques are similar to the differences between the two interpersonal communication models. A detailed algorithm of our proposed method is included in Appendix 6.1.

## 3.4 TRAINING OF THE ENCODER AND DECODER FOR TD

In order to make the interpreting step and interactive communication effective, it is important for the encoder and decoder to learn a reasonable aligned projection between the message space and each of the model's hidden space. This means that given the same input to both teacher and student, encoders of both models need to generate similar messages, and decoded states need to be similar to their original states. To achieve this, besides the $L_{interact}$ used to train student model as well as both student and teacher's encoders and decoders, we introduce following two consistency losses.

**Message space consistency**    Given the same input $x$, student model $M_g$'s encoder $E_g$ and teacher model $M_h$'s encoder $E_h$ will generate similar messages, described below as the message consistency loss between the two messages. This is similar to $L_{feature}$ for feature distillation. Message consistency loss will be used to train the encoder of both models.

$$L_{MC} = d(m_g, m_h) = d(E_g(s_g, e_g), E_h(s_h, e_h))$$

**State space consistency**    Given the same input $x$, the hidden states decoded by two model's message should be consistent with its own hidden states. We introduce the state consistency loss between decoded states and original states below. This is similar to $L_{fitnet}$ in FitNet and is used to train both encoder and decoder of each model.

$$L_{SC} = d(\{s_g; e_g\}, D_g(m_h)) + d(\{s_h; e_h\}, D_h(m_g))$$

The communication encoder and decoder will be co-trained with the training of student model, using the combined loss below:

$$L(x, y, M_g, M_h) = L(y, y_g) + w_1 L_{interact} + w_2 L_{MC} + w_3 L_{SC}$$

Where $L(y, y_g)$ is the groundtruth loss, $w_1, w_2$ and $w_3$ are hyper-parameters of loss weights.

Note that even though we introduce 3 losses ($L_{interact}, L_{MC}$ and $L_{SC}$), the parameters we added for encoder and decoder are comparable to other feature distillation methods. During training, we freeze teacher model and only update student model and the encoder and decoder of both models. Therefore our method doesn't add notable more weights to learn. However, our distillation process takes more time if the number of iterations of interactive communication is larger than one, due to both teacher and student models need to interpret each other's input multiple times.

| Methods | ML(Dense) | ML(Sparse) | CIFAR10 | CIFAR100 | ImageNet | Avg. |
|---|---|---|---|---|---|---|
| Train from Scratch | (—baseline to calculate relative improvement—) | | | | | |
| **LD** | +0.16% | +1.49% | +0.03% | +1.86% | -0.21% | +0.83% |
| **FD** | +0.29% | +2.68% | -0.12% | -0.08% | -0.14% | +0.66% |
| **FitNet** | +0.81% | +2.19% | -0.09% | +0.53% | +0.29% | +0.93% |
| **Hybrid** | +0.93% | +2.91% | +0.23% | +1.94% | +0.02% | +1.50% |
| Our Method (**TD**) | **+1.34%** | **+3.39%** | **+0.45%** | **+2.41%** | **+2.56%** | **+2.54%** |

Table 1: Relative improvement of different distillation methods compared to a student model without distillation. Detailed results with standard error are shown in Appendix 6.5.

| Methods | ML(Dense) | ML(Sparse) | CIFAR10 | CIFAR100 | ImageNet | Avg. |
|---|---|---|---|---|---|---|
| Train from Scratch | (—baseline to calculate relative improvement—) | | | | | |
| No Interaction | +0.81% | +2.02% | +0.43% | +2.26% | +2.44% | +1.99% |
| 1 iteration | +1.31% | +3.08% | +0.42% | +2.31% | +2.49% | +2.40% |
| >1 iterations | **+1.34%** | **+3.30%** | **+0.45%** | **+2.41%** | **+2.56%** | **+2.52%** |

Table 2: Relative improvement of using interactive communication. Detailed results with standard error are shown in Appendix 6.5.

## 4 EXPERIMENT

In this section, we evaluate whether our proposed method can improve knowledge distillation by introducing the interactive communication process, for the case of distilling pre-trained teacher to smaller models for downstream tasks. The teacher model is trained with (multiple) pre-training task(s) or multiple datasets. And the student model will be trained and evaluated on a much smaller dataset with a specific downstream task.

### 4.1 EXPERIMENT SETUP

**Datasets** We choose multiple widely-used real-world benchmark datasets, including MovieLens (Harper & Konstan, 2015), CIFAR10, CIFAR100 (Krizhevsky, 2009), and ImageNet (Russakovsky et al., 2015). These datasets cover applications of recommendation and image classification. For MovieLens (ML), we split the data by timestamps, so that 90% of the past events will be used to train models evaluated by the 10% of the future events, which is close to the real-world setup. The pretrained task is to predict movie ratings given a user and a movie for all genres of movies. And downstream tasks are movie rating prediction for a specific genre. Teacher is an MLP model and student model has only 1/4 of the neurons for each layer in teacher model. For image classification datasets, we adopt the same setup as Vision Transformer (ViT) (Dosovitskiy et al., 2020), where we pretrain a large ViT teacher model on ImageNet21K and evaluate student models using CIFAR10, CIFAR100 and ImageNet. Teacher is the pre-trained 12-layer ViT-B/32 model, and student model is the same architecture but only has 4 transformer layers. Dataset details are in Appendix 6.2.

**Baseline methods** We compare with four baseline methods: Label Distillation (**LD** (Hinton et al., 2015)), Feature Distillation (**FD** (Heo et al., 2019)), **FitNet** (Romero et al., 2014) and a Hybrid version of Label and Feature Distillation (**Hybrid** (Zhu & Wang, 2021)). Note that most recent KD approaches (such as Beyer et al. (2022), Yang et al. (2022a)) focus on one single application such as image classification or recommendation, and assume teacher and student share similar tasks. We cannot directly apply them in our setup, therefore we compare with the general KD algorithms that can be extended to our use case. For a fair comparison, all methods use the same teacher and student structure. Each of the methods' KD loss weights are tuned. Details about baseline methods and Hyper-parameter tuning are shown in Appendix 6.3.

### 4.2 OVERALL IMPROVEMENT

The overall improvement of our method compared to baseline methods is shown in Table 1. We can see that our method outperforms all baseline methods on all 5 tasks. For MovieLens, we report results on two types of downstream tasks: ML(Dense) is from one genre that has dense data and ML(Sparse) contains 4 genres that are much sparser. We include results on all other genres in Appendix 6.5.

Figure 3: Representation Similarity between teacher (column) and student (row).

### 4.3 ABLATION STUDY: UNDERSTAND THE INTERACTIVE COMMUNICATION

To evaluate the performance of the interactive communication process, we conduct ablation study by adjusting number of iterations for interactive communication. We set the loss weight to be same for each iteration. Results are shown in Table 2. For "No Interaction", we disable interactive communication (no $L_{interact}$) but keep the encoder/decoder with consistency losses ($L_{MC}$ and $L_{SC}$). We use 3 as the maximum number of iterations for MovieLens and 2 for ViT. The best results are achieved with the maximum number of iterations. We didn't explore more iterations since our training time will be $k$ times slower where $k$ is the number of iterations. But in real-world scenarios, the distillation cost can be relatively small when downstream tasks are sparse. We also conduct ablation studies on the consistency losses $L_{MC}$ and $L_{SC}$ to show the importance of aligning the message spaces and decoded states between the two models. The results are shown in Appendix 6.5.

### 4.4 CAST STUDY: BRIDGE THE CAPACITY GAP ON UNDISTILLABLE DOWNSTREAM CLASSES

To show that TD can improve student's performance when there is a huge capacity gap between teacher and student, we conduct case study on ImageNet by measuring the representation similarity between student model and teacher model on downstream classes. Similar to the analysis measuring "undistillable" classes (classes cannot be distilled) due to capacity gap (Zhu et al., 2022), we use the Center Kernel Alignment (CKA) (Kornblith et al., 2019) to measure similarities between representations from teacher ($s_h$ and $e_h$) and student ($s_g$, $e_g$ and the message space $m_g$). For each class, we use 20 examples for the analysis. The results are shown in Figure 3. The higher the value (i.e., the lighter the color) on the diagonal term, the better aligned the two representation spaces are. We can see that the learned message space is more aligned with teacher compared to distillation baseline even on "undistillable" classes (classes with darker color on higher layer representation similarities between teacher and student, i.e., between $e_g$ and $e_h$). And the student models learned with TD are more aligned with teacher's representation on the top layer hidden states ($e_g$ and $e_h$).

## 5 CONCLUSION

In this paper, we propose Talking-model Distillation (TD), a new distillation paradigm that enables model to communicate interactively, for transferring the knowledge from large pre-trained foundation models to efficient downstream models. By adapting interpersonal communication models to KD, we first show that existing KD techniques utilize one-way communication from teacher to student. We then design an interactive communication paradigm where teacher and student can exchange messages. In this paradigm, teacher can pass knowledge to student based on student's encoded messages using downstream tasks. And the interactive communication can have multiple iterations, which enables distilling rich information even when downstream data are sparse. We conduct experiments on multiple benchmark datasets and we show that our proposed method outperforms other distillation baselines on improving student performance by distilling pre-trained teacher.

**Implication and limitation** Our proposed method provides a new way of thinking about how models can transfer knowledge and interact with each other. Even though we show early results of interactive communication can further improve model performance for knowledge distillation, the gap between teacher and student is still very large. In the same time, we increase the computation resource used in the communication process, even though it is relatively negligible compared to fine-tuning large foundation models. We believe that by spending computation resources in communication, we enable models from different tasks and modalities to learn from each other efficiently. We also didn't fine-tune the teacher, while the results can change with different fine-tuning methods.

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
