# 6 APPENDIX

## 6.1 PROBLEM DEFINICATION, PSEUDO-CODE AND MORE DISCUSSION OF TD

**Problem definition**  Given teacher model $M_h$, downstream dataset $D = \{X, Y\}$. Train a student model $M_g$ that minimize the loss of $\sum_{(x,y) \in D} Loss(y_g = M_g(x), y)$.

We provide the pseudo code of our proposed Talking-model Distillation (TD) method in Algorithm 1. Note that the state consistency loss $L_{SC}$ and message consistency loss $L_{MC}$ will only be applied on $m_g^0$ and $m_h^0$, which require input $x$ being fed to both teacher and student. These two losses are used to train encoders and decoders. They can be disabled in the later training stage. They can also be optional (ablation results shown in Secion 6.5). Without these losses, teacher model doesn't need to access any input data $x$ of downstream tasks.

---

**Algorithm 1:** Pseudo-Code of the proposed TD

---

**Require:** Trained teacher model $M_h$, initialized student model $M_g$ (or a pretrained smaller model). Initialized encoders ($E_g$, $E_h$) and decoders ($D_g$, $D_h$) for both teacher and students. Downstream dataset $D = \{X, Y\}$. $k$ iterations for interactive communication.
1: get $x$ and $y$ from D
2: $L = 0.0$ // Total loss.
3: $y_g, \{s_g; e_g\} = M_g(x)$
4: $m_g^0 = E_g(\{s_g; e_g\})$
5: $y_h, \{s_h; e_h\} = M_h(x)$
6: $m_h^0 = E_h(\{s_h; e_h\})$
7: // Adding state consistency and message consistency loss to train
8: // communication encoders/decoders.
9: $L = L(y, y_g) + w_2 L_{SC}(m_g^0, D_h, m_h^0, D_g) + w_3 L_{MC}(m_g^0, m_h^0)$
10: **for** each iteration i in [0, k] **do**
11:    $\{s_h'; e_h'\} = D_h(m_g^i)$ // Teacher decodes message from student.
12:    $\tilde{e}_h = H_{l_h+1,...,h_h}^h(s_h')$ // Teacher interpreting step.
13:    $m_h^{i+1} = E_g(\{s_h', \tilde{e}_h\})$ // Teacher encodes returned message.
14:    $\{s_g'; e_g'\} = D_g(m_h^{i+1})$ // Student decodes returned message.
15:    $L = L + w_1 L_{interact}(\{s_g; e_g\}, \{s_g'; e_g'\})$ // Interactive communication loss for iteration i.
16:    $\tilde{e}_g = H_{l_g+1,...,h_g}^g(s_g')$ // Student interpreting step for next iteration.
17:    $\{s_g; e_g\} = \{s_g'; \tilde{e}_g\}$ // set student state for next iteration.
18:    $m_g^{i+1} = E_g(\{s_g; e_g\})$ // Student encodes message for next iteration.
19: **end for**
20: Use optimizer to update the model via total loss $L$.

---

Another design choice we made is to make the communication modules exactly the same between teacher and student. This means teacher encoder and student encoder map their states into the same embedding space: both encoders encode all hidden states (i.e., lower layer and higher layer representations); and decoders decode message to all hidden states. However, for distillation, a specialized design might further improve student performance. For example, message from student to teacher only encodes lower layer representation. (Note that message from teacher to student still needs to encode both lower layer and higher layer representations due to student does both distillation and interpreting). In this paper, we intentionally keep the communication operations the same between teacher and student due to the following reasons.

- Simplicity. We want to verify the concept of communication for knowledge distillation (KD) and the effectiveness of introducing interactive communication to KD. Therefore, we adopt the simple design to keep the communication mechanism the same between teacher and student.

- Future work of multi-way communication. Communication can also happen between a group of models besides two models. Therefore, it is straight-forward to extend the current

communication paradigm (both teacher and student adopt same communication mechanism) to multi-way communication, e.g., multiple teachers, and/or multiple students.

- Future work of multi-way transfer learning. The communication algorithm we propose in this paper can be applied not only in knowledge distillation, but also can be used as a generic way for transferring knowledge among models. Therefore, it can be applied to two models where they can learn from each other, e.g., between a large Vision Transformer Model and a large language model. Therefore, communication can be used for transfer learning with the unified design of communication encoder and decoder.

## 6.2 DATASET DESCRIPTION AND EXPERIMENT SETUP

**MovieLens100k (Harper & Konstan, 2015)**    We use the MovieLens100K dataset included in Tensorflow Dataset [1]. It contains 100K movie ratings. We use 'user_id', 'movie_id', and 'movie_title' and 'movie_genres' as features. We split the data by timestamps. The 90% of the data with earlier timestamps are used for training and 10% of the data with later timestamps are used for evaluation. The time split for evaluation is more realistic than random split for recommendation tasks, since it can capture problems such as user preference shifting overtime as well as cold-starting for new users. We treat training using data from all genres as pre-training task and training on data with movies from specific genres as downstream tasks. We evaluate downstream tasks for 8 most dense genres (with more than 500 evaluation examples), and report the Root Mean Squared Error(RMSE) for rating prediction.

**CIFAR10 (Krizhevsky, 2009)**    We use the CIFAR10 dataset included in Tensorflow Dataset [2]. It contains 60,000 32*32 color images in 10 classes. We use the default training and test split, where there are 50,000 images used for training and 10,000 images used for testing.

**CIFAR100 (Krizhevsky, 2009)**    We use the CIFAR100 dataset included in Tensorflow Dataset [3]. It contains the same 60,000 23*32 color images as CIFAR10, but in 100 classes. We use the same train and test split as CIFAR10.

**ImageNet (Russakovsky et al., 2015)**    We use the ImageNet dataset described in Tensorflow Dataset [4]. It contains 1,281,168 images for training, 50,000 images for validation and 100,000 images for test.

**Teacher model**    For MovieLens tasks, the teacher model is a Multi-layer Perceptron (Gardner & Dorling, 1998), with input dimension 300 (100 for 'user_id', 100 for 'movie_id', and 50 for 'movie_title' using bag-of-words and 50 for 'movie_genres' using bag-of-words). It has two relu layers of 512 units and 256 units. The model size is tuned as hyper-parameters with a upper limit of cost measured by number of flops, and the optimal teacher model size is below the upper limit.

For Image classification tasks, we use Vision Transformer (ViT) (Dosovitskiy et al., 2020) pre-trained on ImageNet21k with available code [5], hyper-parameter setups and checkpoints [6]. The teacher model has 16 transformer layers.

**Student model**    For MovieLens, we set student model size to be 1/4 as teacher model: two relu layers of 128 units and 64 units, where we see significantly quality drop compared to teacher models. For image classification tasks, student model only has 4 transformer layers. We find that using the teacher model's pre-trained weights as initialization for the 4 transformer layers and all other layers results in better and more stable performance compared to random initialization.

---

[1] https://www.tensorflow.org/datasets/catalog/movielens
[2] https://www.tensorflow.org/datasets/catalog/cifar10
[3] https://www.tensorflow.org/datasets/catalog/cifar100
[4] https://www.tensorflow.org/datasets/catalog/imagenet2012
[5] https://github.com/google-research/vision_transformer
[6] https://huggingface.co/google/vit-base-patch32-224-in21k

| Methods | Genre 1 | Genre 2 | Genre 3 | Genre 4 |
|---|---|---|---|---|
| Train from Scratch | $1.0102 \pm 0.0003$ | $1.0884 \pm 0.0004$ | $1.0555 \pm 0.0000$ | $1.0502 \pm 0.0000$ |
| Teacher | $1.0120 \pm 0.0000$ | $1.0854 \pm 0.0000$ | $1.0563 \pm 0.0000$ | $1.0689 \pm 0.0000$ |
| **LD** | $1.0083 \pm 0.0000$ | $1.1029 \pm 0.0000$ | $1.0601 \pm 0.0000$ | $1.0826 \pm 0.0000$ |
| **FD** | $1.0075 \pm 0.0009$ | $1.0997 \pm 0.0030$ | $1.0537 \pm 0.0007$ | $1.0791 \pm 0.0005$ |
| **FitNet** | $1.0018 \pm 0.0000$ | $1.1018 \pm 0.0000$ | $1.0534 \pm 0.0000$ | $1.0849 \pm 0.0000$ |
| **Hybrid** | $1.0015 \pm 0.0008$ | $1.0955 \pm 0.0013$ | $1.0517 \pm 0.0012$ | $\mathbf{1.0737 \pm 0.0014}$ |
| Our Method (**TD**) | $\mathbf{0.9965 \pm 0.0001}$ | $\mathbf{1.0908 \pm 0.0002}$ | $\mathbf{1.0475 \pm 0.0001}$ | $1.0761 \pm 0.0002$ |

| Methods | Genre 5 | Genre 6 | Genre 7 | Genre 8 |
|---|---|---|---|---|
| Train from Scratch | $1.1609 \pm 0.0000$ | $1.1160 \pm 0.0000$ | $1.0038 \pm 0.0000$ | $1.0937 \pm 0.0000$ |
| Teacher | $1.1501 \pm 0.0000$ | $1.1090 \pm 0.0000$ | $1.0193 \pm 0.0000$ | $1.0660 \pm 0.0000$ |
| **LD** | $1.1964 \pm 0.0000$ | $1.1415 \pm 0.0000$ | $1.0088 \pm 0.0000$ | $1.0602 \pm 0.0000$ |
| **FD** | $1.1929 \pm 0.0013$ | $1.1269 \pm 0.0029$ | $1.0024 \pm 0.0000$ | $1.0527 \pm 0.0036$ |
| **FitNet** | $1.1873 \pm 0.0000$ | $1.1260 \pm 0.0000$ | $1.0032 \pm 0.0000$ | $1.0605 \pm 0.0000$ |
| **Hybrid** | $1.1843 \pm 0.0008$ | $1.1215 \pm 0.0028$ | $1.0076 \pm 0.0009$ | $1.0499 \pm 0.0014$ |
| Our Method (**TD**) | $\mathbf{1.1656 \pm 0.0010}$ | $\mathbf{1.1050 \pm 0.0000}$ | $\mathbf{1.0022 \pm 0.0000}$ | $\mathbf{1.0485 \pm 0.0013}$ |

Table 3: RMSE of rating prediction on MovieLens Genre 1 to Genre 8 (Dense to Sparse), compared to baseline methods. **bold** numbers for the best improvement given a certain genre.

**Encoder/Decoder** For encoder and decoder, we use the dense-relu-dense model architecture, with layer norm and dropout. We didn't do extensive hyper-parameter search to choose the best model size, instead, we manually pick relu layer size and message dimension to match the size between teacher and student model. For MovieLens, encoder and decoder have a relu layer with 256 hidden units, and the message dimensionality is 128. For ViT encoder and decoder have a relu layer with 512 hidden units and the message dimensionality is 512.

### 6.3 Hyper-parameter tuning

**Model parameter** For the teacher model on MovieLens, we tune the model size with a upper limit of cost along with learning rate, dropout rate and number of train steps. We don't tune the size of student model, but tune student model's learning rate, dropout rate and number of train steps. For ViT, we use the reported hyper-parameter setup (Dosovitskiy et al., 2020), with fine-tuning steps set to 20000.

**Baseline methods** For each of the baseline methods, we tune their KD loss weight combined with the groundtruch loss weight. For Label Distillation (Hinton et al., 2015) (**LD**) it is the weight of $L_{logit}$, For Feature Distillation (Heo et al., 2019) (**FD**), it is the weight of $L_{feature}$. For **FitNet** (Romero et al., 2014), it is the weight of $L_{fitnet}$. And for Hybrid Distillation (Zhu & Wang, 2021) (**Hybrid**), we tune the weights of overall $L_{logit}$ and $L_{feature}$ and report the best results.

**Our method** We tune the three weights $w_1$, $w_2$ and $w_3$ for our method, which corresponds to the weight of $L_{interact}$, $L_{SC}$ and $L_{MC}$. We also tune the number of iterations for interactive communication. For MovieLens, it is 0, 1, 2 or 3. And for ViT it is 0, 1 or 2. We report the results with different iteration numbers in our ablation study in Section 4.3.

### 6.4 Computation resources

The training of MovieLens can be done on a CPU machine with less than 12 hours for all methods. And the finetuning of ViT models runs on a 4-chip TPU, where all methods finish fine-tuning in 12 hours.

### 6.5 Additional experimental results

In this subsection, we include detailed experimental results. For experiments on MovieLens, both teacher and student models are random initialized. For each result, we run the same setup 5 times and

| Methods | CIFAR10 | CIFAR100 | ImageNet |
|---|---|---|---|
| No Distillation | 0.93678 | 0.74764 | 0.48683 |
| **LD** | 0.93709 | 0.76162 | 0.48576 |
| **FD** | 0.93565 | 0.74702 | 0.48615 |
| **FitNet** | 0.93586 | 0.75164 | 0.48828 |
| **Hybrid** | 0.93894 | 0.76213 | 0.48691 |
| Our Method (**TD**) | **0.94100** | **0.76562** | **0.49930** |

Table 4: Accuracy of image classification tasks, compared to baseline methods. **bold** numbers for the best results on a dataset.

| Methods | Genre 1 | Genre 2 | Genre 3 | Genre 4 |
|---|---|---|---|---|
| No Interactions | $1.0018 \pm 0.0002$ | $1.1016 \pm 0.0002$ | $\mathbf{1.0475 \pm 0.0001}$ | $1.0767 \pm 0.0000$ |
| 1 iteration | $0.9971 \pm 0.0003$ | $\mathbf{1.0908 \pm 0.0002}$ | $1.0502 \pm 0.0007$ | $1.0783 \pm 0.0002$ |
| 2 to 3 iterations | $\mathbf{0.9965 \pm 0.0001}$ | $1.0921 \pm 0.0005$ | $1.0508 \pm 0.0002$ | $\mathbf{1.0761 \pm 0.0002}$ |

| Methods | Genre 5 | Genre 6 | Genre 7 | Genre 8 |
|---|---|---|---|---|
| No Interactions | $1.1864 \pm 0.0014$ | $1.1249 \pm 0.0000$ | $\mathbf{1.0022 \pm 0.0000}$ | $1.0656 \pm 0.0000$ |
| 1 iteration | $1.1667 \pm 0.0008$ | $1.1134 \pm 0.0008$ | $1.0033 \pm 0.0002$ | $1.0496 \pm 0.0026$ |
| 2 to 3 iterations | $\mathbf{1.1663 \pm 0.0016}$ | $\mathbf{1.1050 \pm 0.0000}$ | $1.0039 \pm 0.0004$ | $\mathbf{1.0495 \pm 0.0004}$ |

Table 5: RMSE of rating prediction on MovieLens Genre 1 to Genre 8 (Dense to Sparse), with different number of iterations for interactive communication. **bold** numbers for the best results given a certain genre.

report the mean RMSE with standard error. For experiments using image classification tasks, teacher is pre-trained ViT and students are initialized using the learned weights from pre-trained teacher (only the first four transformer layers), therefore the results have low variance and we only run each setup once due to the limit of computation resources.

**MovieLens100k per genre results**   Results (RMSE, lower is better) on MoiveLens are shown in Table 3. From where we can see that our method outperforms baseline methods on 7 of the 8 genres. We also include the results of the teacher model, which is trained on all genres. The teacher model is not fine-tuned to each downstream genre, and different genres can have very different data distributions. Therefore, in some genres a model trained from scratch is better than the teacher model. And distillation from teacher to student could even hurt the student's performance for some genres. This real challenge in recommendation tasks and many other downstream applications inspires us to design the interactive communication process so that knowledge aligned with downstream tasks can be transferred effectively. We can see that our method, though does not improve the student model on some specific genres, can out-perform both teacher and student on most genres.

**Vision Transformer results**   Results (classification accuracy, higher is better) for image classification tasks are shown in Table 4. We can see that our method outperforms baseline methods on all downstream tasks. Our improvement is most significant on ImageNet, which is a much more difficult task compared to CIFAR10 and CIFAR100. Note that the pre-trained teacher cannot be directly applied to downstream tasks, due to classification label mismatch, so we don't report teacher's results. However, the fine-tuned results can be found in the ViT paper (Dosovitskiy et al., 2020) (0.98, 0.92, 0.81 for CIFAR10, CIFAR100 and ImageNet). We can see that there is still a huge gap between teacher and student.

We want to point out that in this paper we don't discuss the upper limit of the student model nor try to close the gap between teacher and student. In our case, we expect the student with 4 transformer layers to perform much worse than the teacher with 12 transformer layers. We want to verify that by using the proposed interactive communication process, we can transfer more useful knowledge from a powerful pre-trained foundation model to much smaller models for downstream applications, compared to existing KD baseline methods.

| Methods | CIFAR10 | CIFAR100 | ImageNet |
|---|---|---|---|
| No Interactions | 0.94089 | 0.76460 | 0.49873 |
| 1 iteration | 0.94069 | 0.76511 | 0.49893 |
| 2 iterations | **0.94100** | **0.76562** | **0.49930** |

Table 6: Accuracy of image classification tasks, with different number of iterations for interactive communication. **bold** numbers for the best results on a dataset.

| Methods | Genre 1 | Genre 2 | Genre 3 | Genre 4 |
|---|---|---|---|---|
| No $L_{MC}$ | **0.9965 ± 0.0001** | **1.0908 ± 0.0003** | 1.0481 ± 0.0010 | 1.0767 ± 0.0003 |
| No $L_{SC}$ | 0.9968 ± 0.0003 | 1.0916 ± 0.0004 | 1.0501 ± 0.0002 | 1.0763 ± 0.0002 |
| Our Method(**TD**) | **0.9965 ± 0.0001** | **1.0908 ± 0.0002** | **1.0475 ± 0.0001** | **1.0761 ± 0.0002** |

| Methods | Genre 5 | Genre 6 | Genre 7 | Genre 8 |
|---|---|---|---|---|
| No $L_{MC}$ | 1.1683 ± 0.0007 | 1.1104 ± 0.0007 | 1.0035 ± 0.0000 | **1.0485 ± 0.0013** |
| No $L_{SC}$ | 1.1658 ± 0.0012 | 1.1058 ± 0.0002 | 1.0034 ± 0.0001 | 1.0494 ± 0.0004 |
| Our Method(**TD**) | **1.1656 ± 0.0010** | **1.1050 ± 0.0000** | **1.0022 ± 0.0000** | **1.0485 ± 0.0013** |

Table 7: RMSE of rating prediction on MovieLens Genre 1 to Genre 8 (Dense to Sparse), ablating different losses. **bold** numbers for the best results given a certain genre.

**Ablation of interactive communication** We evaluate the effectiveness of interactive communication by changing the number of iterations for calculating the interactive communication loss $L_{interact}$. Results on MovieLens are shown in Table 5 and results on image classification are shown in Table 6. We can see that even without interactive communication, our method can outperform some baseline methods. We think this is because the introduction of both $L_{SC}$ and $L_{MC}$ enables better alignment between the student and teacher's hidden states. It can be viewed as a combination of FitNet and feature distillation. And by introducing interactive communication, we further improve the student model.

**Ablation of consistency losses** We also evaluate the importance of the consistency losses we introduced to help training the communication encoder and decoder. Results on MovieLens are shown in Table 7 and results on image classification are shown in Table 9. For MovieLens, we can see that both message consistency loss $L_{MC}$ and state consistency loss $L_{SC}$ are useful for most genres. For ViT, we always add $L_{MC}$ since we observe that without $L_{MC}$ the communication encoder and decoder is hard to train. And we see that $L_{SC}$ improves the model on CIFAR10 and CIFAR100 but not ImageNet. We think applying $L_{interact}$ with multiple iterations can train the communication encoder and decoder reasonable well, therefore the consistency losses may not always be useful on all downstream tasks.

**Adding noises during communication** Inspired by ideas in self-training and semi-supervised learning (Xie et al., 2020), where noise can be added to input or representation to improve the generalization and robustness of knowledge transfer, we also explored the option of adding noise in the interpreting process. Specifically, we add a small Gaussian noise on $s'_h$, which is the decoded lower layer presentation for teacher model to interpret. Results on MovieLens are shown in Table 8 and results on image classification are shown in Table 9. We can see that adding noise can improve performance on some downstream tasks but not all of them.

**Separate training of encoder/decoder** We also explored different ways of improving the learning of communication encoder and decoder. One way is to introduce a ramp-up stage where only these encoders and decoders are trained. To do this, we first train student model a few steps (1000 on MovieLens) and then we freeze the student model and only train both teacher and student's encoders and decoders for another few steps (500 or 1000 on MovieLens). We report the results on MoiveLens in Table 8, where we can see it does not necessarily improve the student model's performance. One reason is that introducing this ramp-up step will relatively reduce the train steps of end-to-end training. Therefore, it requires more tuning on learning rate, train steps to identify improvement with this training schema. To keep the experiment and algorithm design as simple as possible, in our

| Methods | Genre 1 | Genre 2 | Genre 3 | Genre 4 |
|---|---|---|---|---|
| Add Noise | $0.9966 \pm 0.0001$ | $1.0910 \pm 0.0002$ | $1.0511 \pm 0.0003$ | $1.0762 \pm 0.0003$ |
| No Noise | $0.9966 \pm 0.0002$ | $1.0912 \pm 0.0004$ | $1.0475 \pm 0.0001$ | $1.0766 \pm 0.0002$ |
| Train $E_*, D_*$ separately | $1.0008 \pm 0.0002$ | $1.1066 \pm 0.0001$ | $1.0475 \pm 0.0001$ | $1.0812 \pm 0.0000$ |
| Train together | $0.9965 \pm 0.0001$ | $1.0908 \pm 0.0002$ | $1.0508 \pm 0.0002$ | $1.0761 \pm 0.0002$ |
| **Methods** | **Genre 5** | **Genre 6** | **Genre 7** | **Genre 8** |
| Add Noise | $1.1663 \pm 0.0009$ | $1.1052 \pm 0.0001$ | $1.0033 \pm 0.0001$ | $1.0487 \pm 0.0014$ |
| No Noise | $1.1668 \pm 0.0015$ | $1.1051 \pm 0.0002$ | $1.0026 \pm 0.0003$ | $1.0504 \pm 0.0012$ |
| Train $E_*, D_*$ separately | $1.1782 \pm 0.0002$ | $1.1165 \pm 0.0002$ | $1.0022 \pm 0.0000$ | $1.0603 \pm 0.0002$ |
| Train together | $1.1656 \pm 0.0010$ | $1.1050 \pm 0.0000$ | $1.0061 \pm 0.0001$ | $1.0485 \pm 0.0013$ |

Table 8: RMSE of rating prediction on MovieLens Genre 1 to Genre 8 (Dense to Sparse), by adding noise before teacher's interpreting or separately training encoder/decoder.

| Methods | CIFAR10 | CIFAR100 | ImageNet |
|---|---|---|---|
| No $L_{SC}$ | 0.94069 | 0.76398 | 0.49930 |
| Add Noise | 0.94089 | 0.76562 | 0.49917 |
| No Noise | 0.94100 | 0.76511 | 0.49930 |
| Our Method(**TD**) | 0.94100 | 0.76562 | 0.49930 |

Table 9: Accuracy of image classification tasks, ablating $L_{SC}$ or adding noises before teacher's interpreting.

experiments, we train everything (student model, both teacher and student's encoders and decoders) together in a single stage.