# OpenReview forum: "Talking Models: Distill Pre-trained Knowledge to Downstream Models via Interactive Communication"
_ICLR.cc/2024/Conference — Submitted to ICLR 2024_

### Official Review · Reviewer_2VSL · 2023-10-30

**Soundness:** 3 good
**Presentation:** 2 fair
**Contribution:** 3 good
**Rating:** 5
**Confidence:** 4

**Summary:**

In their paper, the authors present a novel technique for knowledge distillation that leverages an interactive communication process. This approach draws inspiration from Osgood-Schramm's two-way communication model and employs communication encoders and decoders. Additionally, the authors introduce three supplementary loss functions to guarantee the desired behavior of the distillation process. To assess the efficacy of their method, they conduct experiments on four different datasets, covering two distinct tasks: movie recommendation and image classification. The results of these experiments demonstrate that this interactive distillation process can lead to performance enhancements.

**Strengths:**

1. The incorporation of Osgood-Schramm's model into the knowledge distillation process is novel and interesting.
2. The introduction of three new loss functions helps to realize the desired interactive distillation process.
3. The paper has a well-crafted structure and easy to follow.

**Weaknesses:**

The paper has several limitations that need to be addressed:

1. **Limited Comparison Baselines:** The study only compares the proposed method with four baseline approaches. To provide a more comprehensive evaluation, it is advisable to consider more advanced knowledge distillation techniques and include a comparison with state-of-the-art models in the field. For instance, [a],

2. **Limited Tasks:** The paper only explores two specific tasks, which may not represent the full spectrum of potential applications for the proposed approach. Expanding the scope of evaluation to cover a broader range of tasks would provide a more robust assessment.

3. **Insufficient Comparison with IAKD:** While the paper introduces a novel approach, it does not adequately differentiate it from Interactive Knowledge Distillation (IAKD). A clear comparison highlighting the advantages and distinctions between the proposed method and IAKD is needed to help readers understand the contribution.

4. **Underwhelming Performance:** The reported performance metrics, such as RMSE and accuracy in Table 3 and 4, do not appear to be competitive when compared to state-of-the-art results. The ablation study also suggests that the new losses (L_MC and L_SC) do not significantly improve performance. For more up-to-date results on the datasets, it is recommended to refer to sources like [RMSE on ML100k](https://paperswithcode.com/sota/collaborative-filtering-on-movielens-100k) and [Cifar-10](https://paperswithcode.com/sota/image-classification-on-cifar-10) to provide a clearer context for your results.

Addressing these issues will help strengthen the paper and provide a more comprehensive and competitive assessment of the proposed approach.

[a] Radhakrishnan, Adityanarayanan, et al. "Transfer learning with kernel methods." Nature Communications 14.1 (2023): 5570.

**Questions:**

1. How dose this approach perform on natural language processing tasks such as text classification, token classification, question answering, etc. Further investigation on these NLP tasks is essential to assess the adaptability and effectiveness of the proposed method in a broader range of applications.
2. How is the approach compared with the most recent knowledge distillation methods? To establish the novelty and competitiveness of the proposed method, it is crucial to benchmark it against recent state-of-the-art knowledge distillation techniques, considering various datasets and evaluation metrics.
3. How do you determine w_1, w_2, w_3? The determination of the weights, namely w_1, w_2, and w_3, is not clearly elucidated in the paper.

---

> ### Author Response · Authors · 2023-11-23
>
> We thank the reviewer for the on-point comments and insightful questions. We address the reviewer’s questions below:
>
> Weaknesses:
> 1. Limited Comparison Baselines
>
> We thank the reviewer for providing an additional baseline method. One thing we want to point out here is that our goal is not to improve knowledge distillation in general, instead, we want to extend knowledge distillation to the case where it can be used to help transfer knowledge efficiently from pre-trained large models to smaller models specific for downstream tasks. Therefore, we don’t compare with many KD baselines, we carefully select the canonical ones from which our method builds upon and can be extended to our application. Since we use intermediate representations, many state-of-the-art feature distillation methods (such as [4]) can be used in our framework by modifying the distance metric d in section 3.3. We carefully examine the paper the reviewer provided. It proposed a very scalable and general method for transfer learning with the kernel method. It seems non-trial to apply this method to our problem setup, which is knowledge distillation between pre-trained teacher model and downstream student.
>
> 2. Limited tasks.
>
> We totally agree that expanding the scope of evaluation with more tasks would provide a more robust assessment. This is also the reason why we picked two datasets from totally different domains, i.e., image classification and recommendation systems. We believe that these two different domains can cover many popular use cases. We also studied natural language tasks, however, the task being formulated as generative language modeling tasks totally changed the paradigm of knowledge distillation. Therefore, we focus on verifying our method on the CV and recsys domains.
>
> 3. Insufficient Comparison with IAKD.
>
> We thank the reviewer for pointing out IAKD. IAKD indeed shares similar motivation as our work. But the key idea of IAKD is very different and cannot be directly applied in our case. IAKD directly swapped blocks of models, which require models to be similar in model architecture. In the IAKD paper, the authors used resnet without a pre-training dataset. In our case, we use ViT pre-trained on ImageNet21k, which is much larger and cannot be directly swapped to students due to layer and shape mismatch.
>
> 4. Underwhelming Performance
>
> We thank the reviewer for pointing out the SOTA results. And we want to emphasize that by using smaller models, it is still not realistic to achieve results better than or close to SOTA. This is actually one of the key discussions we want to raise in our paper (Section 5 and Appendix 6.5). In our humble opinion, we don’t think closing the gap between student and teacher model’s performance (fine-tuned teacher in this case, which is SOTA) is realistic for any distillation techniques. The gap caused by capacity and training resources (training extensively on pre-trained data) cannot be overcomed by simply letting one teach the other, as pointed out by many related works [2], [3]. But we still think it’s hopeful that using large pre-trained models can improve downstream models (compared to without distillation), especially when fine-tuning is not realistic in training (limited resource and access to pre-trained model) and serving (serving cost limitation).
>
> Especially the training from scratch ViT model is not as good as some of the SOTA results, as discussed in [1], ViT models can be relatively hard to train from scratch. However, we want to highlight the relevant improvement that different distillation methods can bring with the help of a large pre-trained ViT model. We believe this is an important use case, because not only is fine-tuning a large pretrained model costly, there will be much more cost when actually serving the model. And in many use cases, the limitation on serving makes it impossible to use a large model hence smaller downstream model is our only choice. In this case, the proper baseline is the training from scratch smaller model, not the fine-tuned teacher model.

---

> ### Author Response · Authors · 2023-11-23
>
> To answer reviewer's questions:
>
> 1. How dose this approach perform on natural language processing tasks?
>
> We actually conducted additional experiments on language modeling tasks using T5 to show by asking questions to T5 Large, we can significantly improve the T5 small student on downstream benchmarks such as SuperGLUE. However, we decided not to include this result due to the design of encoding/decoding (through text prompt) and the communication process is entirely different to our current method, even though the main idea is the same. We want to focus on the general concept of interactive communication in this paper.
>
> 2. How is the approach compared with the most recent knowledge distillation methods?
>
> As discussed above, we compared with  the canonical ones from which our method builds upon and can be extended to our application. However, many most recent knowledge distillation for specific domains can be extended and integrated into our interactive communication framework.
>
>
> 3. How do you determine w_1, w_2, w_3?
>
> We conducted hyper-parameter search on our method and baseline methods, the details of hyper-paramter search and results are discussed in Appendix 6.3 and 6.5.
>
> [1] https://arxiv.org/abs/2106.01548
>
> [2] https://openreview.net/forum?id=q6bZruC3dWJ
>
> [3] https://openreview.net/pdf?id=0ltDq6SjrfW
>
> [4]  https://openreview.net/pdf?id=157Usp_kbi

---

### Official Review · Reviewer_93jL · 2023-10-31

**Soundness:** 2 fair
**Presentation:** 3 good
**Contribution:** 3 good
**Rating:** 6
**Confidence:** 4

**Summary:**

This paper proposes a knowledge distillation approach for knowledge transfer from large scale pre-trained foundation models to specific downstream tasks. The approach leverages the design of encoder and decoder for better communication and to shorten the gap between teacher and student models。

**Strengths:**

1. The topic of distilling pre-trained knowledge to benefit the downstream tasks is important and practical.
2. The solution is building up interactive communication between teacher and student models by encoder and decoder is novel and quite interesting.
3. The results look reasonable.
4. The paper is clearly written and well presented.

**Weaknesses:**

1. The experiments on movie prediction only cover a narrow scope, and the teacher/student tasks are quite similar with student task is to predict movie from one genre. The results could be more convincing if more varied tasks are involved, and if the "gap" between teacher and student is larger.
2. The approach makes sense but quite straightforward by adding teacher receiving messages. It's worth more discussion on insights of this effect to the teacher model (if not frozen).

**Questions:**

Same as above.

---

> ### Author Response · Authors · 2023-11-23
>
> We thank the reviewer for the careful examination and insightful comments for our paper. We address the reviewer’s comments below:
> Weaknesses
>
> 1. The experiments on movie prediction only cover a narrow scope, and the teacher/student tasks are quite similar with student task is to predict movie from one genre. The results could be more convincing if more varied tasks are involved, and if the "gap" between teacher and student is larger.
>
> We agree with the reviewer that our method can be further verified with more distribution and task differences. In fact, this is also one of the reasons we choose movielens to be one of our evaluation datasets. The user behaviors on movies from different genres can be entirely different, especially on genres with very few user interactions, such as documentary, it can have totally different user behaviors patterns compared to popular genres such as comedy. We intentionally split the genres to two categories, i.e., dense and sparse during our evaluation. We reported their results separately, where we can see improvement of our methods on both categories. Then on image classification, the distribution between CIFAR10, CIFAR100 and ImageNet can also be very different. We believe that our experiments covered different scenarios. We think it can definitely be improved via experiments in more extreme case, e.g., distilling language modeling teacher to movie lens students, we included this as one of the scenarios we want to verify in our future work.
>
> 2. The approach makes sense but quite straightforward by adding teacher receiving messages. It's worth more discussion on insights of this effect to the teacher model (if not frozen).
>
> We thank the reviewer for the great insights. It is indeed very interesting to discuss the effect on the teacher model if not frozen. In our paper, we conduct analysis by comparing the representation between teacher and student on input images with different classes, where we have shown by introducing the encoder/decoder, we can better align their representations. It can be different when we enable fine-tuning. This is indeed one of our future work, however, some early results suggest fine-tuning doesn’t always align the representations between teacher and student. We added some discussion in our future work section.

---

### Official Review · Reviewer_yycb · 2023-11-04

**Soundness:** 1 poor
**Presentation:** 1 poor
**Contribution:** 2 fair
**Rating:** 5
**Confidence:** 2

**Summary:**

The paper interprets the standard knowledge distillation as one-way communication and proposes an interactive communication method to distill knowledge from large models to small models.

**Strengths:**

The idea of interactive communication between the teacher model and student model is interesting and novel. The introduction and Related work sections are very clear.

**Weaknesses:**

1. The idea seems novel and interesting, but direct evidence is lacked to support its advantages. The analogy of personal communication, though also interesting, is not enough to explain why the proposed method works. We know that the two models are interacting with each other, but with the concrete communication method, it is hard to say that they are actually "talking" to each other like two persons as hypothesized in Introduction. We actually don't know why the proposed method works. In fact, it is hard to understand the rational of the three proposed loss L_{interact}, L_{MC} and L_{SC}. For example, why should the messages of the teacher and the student be consistent (L_{MC}), considering that they are produced by the two models sequentially?

In addition, the two additional encoders and two additional decoders can account for most unaligned factors governed by the last three terms in the last equation in page 7 because these four modules are learnable. Then how much internal knowledge of the teacher model could be transferred to the student model by modifying its parameters?

I doubt that the performance improvement largely comes from the four additional modules as they bring more parameters. A desirable baseline approach for comparison is a knowledge distillation method (such as the one illustrated in Fig 1 left) with some additional modules (e.g., adding some modules between the student and teacher).

2. The experiments are not enough to support the advantage of the proposed method. The compared methods are quite old. It is stated that: Note that most recent KD approaches (such as Beyer et al. (2022), Yang et al. (2022a)) focus on one single application such
as image classification or recommendation, and assume teacher and student share similar tasks. This does not make much sense because the authors could compare with those recent approaches on (same) single applications individually.

3. The presentation is poor. The paper introduces too many notations without a clear rule, in other words, the notations seem to be introduced in an arbitrary manner. For example, the subscripts g and h are used to indicate the student and the teacher, respectively. But in other places, h is used to indicate higher hidden layers of a neural network. This leads to weird notations such as H_{h_{h}}^h, a total of four h's! It is hard to get the meaning of a notation by looking at it. I spent a difficult time in reading the paper. In my opinion, many notations and equations are actually unnecessary. The proposed method is simple, and there is really no need to use such a complicated and tedious manner to describe it.

4. Some technical details are missing. For example, each iteration between the teacher and the student will result in three additional losses (the last three terms in the last equation in page 7). Then with N iterations, does it mean that we need add 3N additional losses? If yes, how should we set the weighting factors? For another example, the method part introduces an encoder-decoder pair for both student and teacher, but in Appendix, only two modules are described. Is the encoder-decoder pair shared by the teacher and the student?

**Questions:**

The first two points listed above.

---

> ### Author Response · Authors · 2023-11-23
>
> We thank the reviewer for the comments, careful examination of our paper, and understanding of our key insights. And we address the reviewer’s comments below:
>
> Weaknesses:
> 1. direct evidence is lacking to support its advantages.  For example, why should the messages of the teacher and the student be consistent (L_{MC}), considering that they are produced by the two models sequentially? I doubt that the performance improvement largely comes from the four additional modules as they bring more parameters.
>
> We thank the reviewer for the question. We provided some evidence that our interactive communication method works better than other one-way communication based methods by some case study included in Section 4.4. We showed that the ``undistillable’’ class between teacher and student can be distillable after the training of encoder and decoder. We believe the training of encoder and decoder with the consistency loss can provide two benefits: (1) Prevent inefficient knowledge transfer from domain shift between pre-trained data and downstream tasks. (2) Enable the teacher to understand what information students can provide, because the student cannot get complex representation from the input as the teacher due to capacity gap, this will allow the teacher to teach with limited representation suitable for the student’s capacity.
> We provided a detailed discussion on the current design regarding the reviewer’s questions in Appendix 6.1 (for design simplicity and future work). To verify our explanation and insights, we also show that (figure 3) using interactive communication, we can bridge the capacity gap between teacher and student for “undistillable classes”[1].
>
> For the question on performance improved due to additional parameters: our method didn’t add significant computation costs compared to existing KD methods. Costs added by encoder and decoder are comparable to feature distillation methods. But our costs are (k-1 times) higher when we are doing interactive communication in k iterations. Besides discussion in section 4.3, we will add a quantitative analysis in our revision.
>
> 2. The experiments are not enough to support the advantage of the proposed method.
>
> One thing we want to highlight is that our paper is not targeted at improving knowledge distillation in general. Our focus is to extend knowledge distillation to the application of efficient knowledge transfer from pre-trained large models to smaller models for downstream tasks. Therefore, different from general KD methods, our proposed method considers two additional challenges: (1) the teacher model is trained differently using pre-trained tasks. (2) fine-tuning the teacher to arbitrary downstream tasks is not practical. We believe this is a very important application as pre-trained foundation models have shown great advantages in different domains such as NLP and CV, and directly using such a large model is not realistic as fine-tuning is not practical, nor serving the fine-tuned large model.
>
> Therefore, we don’t compare our methods to the KD algorithms which cannot be used in our application without non-trivial modification. Another reason is that since our application is on efficient knowledge transfer from large pre-trained models to smaller downstream models, we don’t want our method to be tied to a specific domain (e.g., CV), instead, we evaluate our method on various domains including recommendation system and CV. Therefore, many KD methods targeting a single domain, e.g., CV, or NLP, cannot be directly used to compare with our method.
>
> However, with our proposed interactive communication framework, we can integrate many feature distillation methods to further improve KD in different domains. As an example, the two papers the reviewer pointed out can be very useful to improve KD for ViT and they can be integrated by modifying the distance function d in section 3.3 (for example, mimincing and generating in ViTKD can be used to improve state and message consistency losses).
>
> 3. The presentation is poor.
>
> We will make significant improvements to make our presentation clear in our revision.
>
> 4. Some technical details are missing.
>
> We thank the reviewer for pointing this out! We include our experimental designs in Appendix 6.2 and 6.3. For the losses introduced in interaction, only the interactive loss “L_{interact}” are added multiple times with equal loss. We made this clear in our revision.
>
> [1] https://openreview.net/forum?id=q6bZruC3dWJ

---

### Official Review · Reviewer_njLH · 2023-11-08

**Soundness:** 2 fair
**Presentation:** 3 good
**Contribution:** 3 good
**Rating:** 6
**Confidence:** 3

**Summary:**

The paper presents a new distillation framework that aims to distill knowledge from a pretrained foundation model to a smaller model for downstream tasks. The method is inspired by an interactive communication model, and instantiated by an encoder-decoder architecture. Such a design allows transferring knowledge adaptively based on student model capacities and handling different task distributions. Experiments on vision and recommendation are conducted to verify its effectiveness.

**Strengths:**

1. I haven't kept up with recent developments in KD, both problem setting and the proposed algorithm appear to be novel given the context provided in the paper.
2. Distillation across different tasks or distributions is challenging problem, yet the proposed model performs well in both vision and recommendation applications.
3. The paper is generally well written, the idea is easy to follow. The analogy between KD and communication models is interesting. It provides a unified view of existing KD approaches and is a clever choice for motivational purpose.

**Weaknesses:**

1. While the method intuitively makes sense and I understand the paper is centered on applications, it would be nice to make the paper more formal, e.g. by defining different task distributions and the problem you are to tackle.
2. The link between the method and different task distributions does not seem very clear (partially also due to a lack of formality). Particularly, I still do not fully understand why extending KD to a two-way interactive communication process is relevant solving distribution shift.
3. In terms of writing, I do not find the first half of the paper (section 1 and 2) very informative. I think empirical studies in 4.2-4.4 are especially useful for justifying such type of approach, but regrettably they are not highlighted in the main paper.

Minors: better use vector graphics such pdf rather than bitmap for figures.

**Questions:**

1. How are $l_g$ and $h_g$ chosen? There are also many other hyperparameters, how are they tuned?
2. Can you provide more insights on the question in weakness 2?
3. How is the approach related with foundation models, as teacher models are just some pretrained models, the same as standard KD setting?
4. How distribution shifts are reflected in experiments?
5. Can you discuss the connection with existing cross-task KD approaches?

---

> ### Author Response · Authors · 2023-11-23
>
> We thank the reviewer for the thorough comments and insightful questions, especially pointing out the weak connection between our method and distribution shifts. We address the reviewer’s comments below:
>
> Weaknesses:
> 1. it would be nice to make the paper more formal, e.g. by defining different task distributions and the problem you are to tackle.
>
> Thanks for the suggestion, we added a formal definition of our problem in our Appendix, along with the formal description of the algorithm. However, the definition of task/data distribution seems hard to be added in the formal definition since we only empirically verify our hypothesis on the distribution shift between pre-training and downstream datasets. The theoretical quantification of the differences is still challenging.
>
> 2. The link between the method and different task distributions does not seem very clear.
>
> The connection between our method and task distribution lies in the pre-training fine-tuning paradigm of large foundation models. In this paper, we are not improving the knowledge distillation algorithm in general, but extending knowledge distillation to a very specific use-case: leveraging the knowledge of pre-trained large models to teach smaller student models for downstream tasks, without extensive or at-all fine-tuning. We believe this is a very important application as pre-trained foundation models have shown great advantages in different domains such as NLP and CV. In this case, the teacher cannot adapt to the downstream tasks’ distribution, which is often different from the pre-training distribution. Therefore, we propose to use the encoder and decoder to help the teacher learn the downstream tasks’ distributions without modifying the teacher model. We strengthened our introduction to make this clear.
>
> 3. I do not find the first half of the paper (section 1 and 2) very informative. I think empirical studies in 4.2-4.4 are especially useful for justifying such type of approach, but regrettably they are not highlighted in the main paper.
>
> Thanks for pointing this out! Following your suggestion, we modified our paper to mention our empirical study results in our introduction to help justify our approach.
>
> Minors: better use vector graphics such as pdf rather than bitmap for figures.
> Thanks for the suggestion, we replaced our figures to pdfs.

---

> ### Author Response · Authors · 2023-11-23
>
> To answer reviewer's questions:
> 1. How is hg and lg chosen? There are also many other hyperparameters, how are they tuned?
>
> We don’t tune h_g and l_g, instead, l_g is just after the first input projection layer and h_g is before the last classification layer. Therefore, we enable teacher and student to align their input and label space via the encoder and decoder. We totally agree with the review that deciding which layers are lower and higher is tricky. The point we want to highlight is that we don’t tune this as hyper-parameters. We believe (and use the setup) that the lower should be as low as possible and higher can be as high as possible. Because this is to solve the challenge of domain and task differences between teacher and student. The question of what proper layer and techniques to distill can be partially solved using existing `feature distillation` techniques, as they can be extended into our framework by modifying the distance function d in the equations in section 3.3. There are discussions in some recent feature distillation works [1] and [2] on which layers and losses to use for feature distillation. We will add more discussion and clarification in our revision.
>
> As for other hyper-parameters, we performed grid search of some of hyper-parameters such as loss weights for our method and baseline methods. We included the detailed hyper-parameter tuning in Appendix 6.3.
>
> 2. Can you provide more insights on the question in weakness 2?
>
> Discussed above.
>
> 3. How is the approach related with foundation models, as teacher models are just some pretrained models, the same as standard KD setting?
>
> The key difference here is that we assume fine-tuning teachers for downstream tasks is costly. And also the pretraining dataset can be very general and broad covering various distributions but the downstream tasks are relatively specialized and different from the pre-training dataset and tasks. We believe this differentiates our problem setup with the standard KD setting.
>
> 4. How distribution shifts are reflected in experiments?
>
> We designed our experiments as the pre-trained teacher model is not fine-tuned for downstream tasks. For example, on Movielens, the pretrained model sees data from user ratings on all movie genres while the student is trained on a specific genre. We can see that students for different genres benefit differently from the teacher model using different baseline methods.
>
> 5. Can you discuss the connection with existing cross-task KD approaches?
>
> Thanks the reviewer for pointing out this related literature. By comparing our method with some recent cross-task KDD approaches [3,4,5], our proposed method and problem setup have the following differences:
> (1) The data for training the teacher and student is largely different. Existing cross-task KD don’t assume the distribution differences on the input side between pre-training data and fine-tuning data.
> (2) Limited ability in modifying the teacher.
> (3) Method: These work adopt typical or modified feature distillation paradigm, where our method introduced the inter-active communication paradigm.
>
>
> [1] https://arxiv.org/pdf/2209.02432.pdf
>
> [2] https://arxiv.org/pdf/2104.09044.pdf
>
> [3] Ye et al., Distilling Cross-Task Knowledge via Relationship Matching
>
> [4] Li et al., Prototype-guided Cross-task Knowledge Distillation for Large-scale Models
>
> [5] Yang et al., Cross-Task Knowledge Distillation in Multi-Task Recommendation

---

### Author Response · Authors · 2023-11-23
**Thank you for reviewing!**

We want to thank all the reviewers for the careful review and insightful comments. We address each reviewer’s comments and answer the questions below.

We agree with the reviewer's assessment on the novelty and contribution of this paper. We believe our proposed interactive communication process is a new way to extend knowledge distillation to more exciting applications.

We also want to provide explanations on our experimental setup to address some concerns related to baseline and improvements: Our paper does not focus on improving the general Knowledge Distillation (KD) algorithm. Instead, we focus on extending knowledge distillation to a very specific use-case: leverage the knowledge of pre-trained large models to teach smaller student models for downstream tasks. We believe this is a very important application as pre-trained foundation models have shown great advantages in different domains such as NLP and CV. Therefore, we consider the challenges of: (1) domain and task differences between teacher and student and (2) the large capacity gap without the ability of modifying the teacher.

These challenges made many advanced KD techniques difficult to be modified and applied in our case. To tackle these challenges, we propose interactive communication, where the teacher directly provides “responses” based on student’s “questions”, via intermediate representations. The framework itself can be integrated with many feature distillation techniques, by modifying the actual distillation loss (function d) described in section 3.3.

Because our goal is to improve the smaller downstream models, as usually directly using a large model for serving is not practical, we don’t aim to make the smaller downstream model perform as good as the large pre-trained foundation model. Instead, we show that our proposed method can efficiently transfer most of the knowledge from the pretrained model to downstream tasks, compared to the reasonable modification of existing KD methods without interactive communication.

We are thankful for the very insightful discussion with the reviewers. We hope our response could clarify our paper and solve reviewers’ concern (especially on experiments, e.g., baseline methods, improvement, and hyper-parameter tuning), and glad to have more discussion on other interesting questions and comments.

---

### Comment · Area_Chair_cFaF · 2023-11-23
**From AC at the end of rebuttal: Reviewer response required**

Dear Reviewers,

Thanks for your time and commitment to the ICLR 2024 review process.

As we approach the conclusion of the author-reviewer discussion period (Wednesday, Nov 22nd, AOE), I kindly urge those who haven't engaged with the authors' dedicated rebuttal to please take a moment to review their response and share your feedback, regardless of whether it alters your opinion of the paper.

Your feedback is essential to a thorough assessment of the submission.

Best regards,

AC

---

### Meta-Review · Area_Chair_cFaF · 2023-12-11

**Metareview:**

This paper aims to improve knowledge distillation so that the shifts in model capacities and in data distributions between upstream and downstream can be addressed through the "interactive communication." While the ambition is attracting, the solution presented is unsatisfactory. Reviewers held concerns on how the the proposed method can address the distribution shift problem because there are no direct relation and evidence. The motivation of how communication can help out solving the targeted problems remains vague, in that the analogy between the proposed method and interactive communication is farfetched, and in that the "talking" mechanism has not been well established. The evaluation is not solid enough since many relevant KD methods were left out for comparison, while the task variability and diversity were not sufficient in the context of large pre-training models---these large models are usually powerful to solve many tasks and transfer very well. Even so, the approach's performance is not overwhelming. There are other concerns regarding the subpar presentation quality of the paper. The author's rebuttal failed to convince reviewers in general. In summary, the paper needs to undergo major revisions.

**Justification For Why Not Higher Score:**

The main story of "interactive communication" and "talking mechanism" were not fully established by the proposed encoder-decoder design. Subpar empirical performance even by omitting many relevant KD baselines and on limited variety of tasks. Unsatisfactory writing.

**Justification For Why Not Lower Score:**

N/A

---

### Decision · Program_Chairs · 2024-01-16

Reject